# Glyphosate-Induced Abscisic Acid Accumulation Causes Male Sterility in Sea Island Cotton

**DOI:** 10.3390/plants12051058

**Published:** 2023-02-27

**Authors:** Guoli Qin, Nan Zhao, Weiran Wang, Meng Wang, Jiahui Zhu, Jing Yang, Feng Lin, Xinglei Huang, Yanhui Zhang, Ling Min, Guodong Chen, Jie Kong

**Affiliations:** 1College of Agriculture, Tarim University, Alar 843300, China; 2Institute of Cash Crops, Xinjiang Academy of Agricultural Sciences, Urumqi 830091, China; 3College of Agronomy and Biotechnology, China Agricultusral University, Beijing 100000, China; 4Key Laboratory of Crop Genetic Improvement, Huazhong Agricultural University, Wuhan 430070, China; 5College of Grassland Sciences, Xinjiang Agricultural University, Urumqi 830052, China; 6The State Key Laboratory of Genetic Improvement and Germplasm Innovation of Crop Resistance in Arid Desert Regions (Preparation), Urumqi 830052, China

**Keywords:** sea Island cotton, male sterility, glyphosate, abscisic acid response

## Abstract

Sea Island cotton is the best quality tetraploid cultivated cotton in the world, in terms of fiber quality. Glyphosate is a widely used herbicide in cotton production, and the improper use of herbicides has led to pollen abortion in sea island cotton and, consequently, to a dramatic decrease in yield; however, the mechanism remains unclear. In this study, different concentrations (0, 3.75, 7.5, 15, and 30 g/L) of glyphosate were applied to *CP4-EPSPS* transgenic sea island cotton Xinchang 5 in 2021 and 2022 at Korla, with 15 g/L glyphosate chosen as the suitable concentration. By comparing the paraffin sections of 2–24 mm anthers in the 15 g/L glyphosate treatment group with those in the water control group, we showed that the key period of anther abortion after glyphosate treatment was the formation and development of tetrads, which corresponded to 8–9 mm buds. Transcriptome sequencing analysis of the treated and control anthers revealed a significant enrichment of differentially expressed genes in phytohormone-related pathways, in particular abscisic acid response and regulation pathways. Additionally, after treatment with 15 g/L of glyphosate, there was a significant increase in the amount of abscisic acid in the anthers in the 8–9 mm buds. Further analysis of the differential expression of abscisic acid response and regulatory genes, an abscisic acid response gene *GbTCP14* (*Gbar_A11G003090*) was identified, which was significantly upregulated in buds with 15 g/L glyphosate treatment than the control, and it could be a key candidate gene for the subsequent research involving male sterility induced by glyphosate in sea island cotton.

## 1. Introduction

Sea Island cotton is known for its excellent fiber quality, with its fiber being one of the key raw materials for international high-end textiles and apparel [1]. In China, Xinjiang is the only production area of Sea Island cotton due to its unique climatic resources and light and heat conditions [2]. Chemical weed control is currently one of the most common methods in the field [3]. The improper application of chemical herbicides on a large scale can cause the emergence of male sterility, the loss of bolls, and a decrease in yield [4].

As the most widely used herbicide in production, glyphosate is a competitive inhibitor of the 5-enolpyruvate manganate 3-phosphate synthase (EPSPS), present in the plant metabolic pathway, which is required for the synthesis of the aromatic amino acids: tryptophan, tyrosine, and phenylalanine [5]. Viator et al. treated cotton with different concentrations of glyphosate and found that the boll yield decreased as the accumulation of glyphosate increased [6]. Chen et al. found that, after glyphosate treatment, the *EPSPS* protein expression was lower in anthers compared to the other tissues [7]. Yasuor et al. investigated glyphosate-treated anthers and found that anthers did not dehisce, and pollen activity was reduced [8]. Additionally, glyphosate had a direct effect on the content of carotenoid in yellow nutsedge [9], which in turn affected the biosynthesis of abscisic acid [10]. In Pennisetum, glyphosate increased the content of endogenous abscisic acid [11]. Studies on male abortion, induced by glyphosate, in upland cotton have been reported [12], although not yet in sea island cotton, and the specific molecular mechanism is uncertain.

Male sterility, primarily anther abnormity, is the combined result of multiple metabolic pathways regulated by several genes, including transcription factors. Of these, plant hormones, particularly abscisic acid, play an irreplaceable role in the development of reproductive organs. In magnolia, stamen degeneration was associated with abnormal ratios of auxin, abscisic acid, and gibberellin [13]. In wheat, the decrease in auxin, abscisic acid, and gibberellin, and the rise in zeatin led to pollen abortion [14]. In barley, Kasembe improved the sterility of male sterility mutants by exogenous gibberellin spraying [15]. In cucumber, gibberellin promoted the development of male flowers [16]. In peanuts, calcium deficiency caused the disorder of phytohormones, for example, the rise of auxin, leading to sterility in peanuts [17]. In tomatoes, the downregulation of *PIN-FORMED*, a gene encoding the auxin transporter protein, led to pollen abortion in tomatoes [18]. The higher concentrations of abscisic acid can lead to abnormal anthers [19]. Abscisic acid also causes pistil abortion in Japanese apricot [20]. In addition, transcription factors, such as TCP, play crucial roles in male sterility [21]. The deletion of the *TCP4* gene resulted in defective stamens in the anthers. In lonicera, the TPC family mainly regulates the symmetry of flower development, while the upregulated expression of *CYC* inhibited the growth of stamens, and led to abortion [22]. In maize, the members of the TCP family promote the production of female inflorescences [23].

To investigate the glyphosate concentration responsible for causing male sterility in Sea Island cotton and the corresponding mechanism, this study examined the boll number, flower morphology, buds, and anthers of Sea Island cotton treated with different concentrations of glyphosate. In addition, we investigated the molecular expression of the shikimic acid pathway induced by glyphosate and detected the metabolite content to explore the effect of glyphosate on the shikimic acid metabolism pathway. Through cytological observations, we determined the critical period for glyphosate-induced anther abortion. Transcriptome sequencing analysis has yielded the key genes responsible for male sterility in Sea Island cotton. Finally, changes in the levels of key phytohormones associated with sterility were detected. This study provides methodological guidance on the scientific use of chemical herbicides and lays a solid theoretical foundation for the future expansion of Sea Island cotton production.

## 2. Results

### 2.1. Glyphosate Causes Yield Loss by Inducing Male Sterility in Sea Island Cotton

After 15 days of treatment with glyphosate spray, the corolla of the treated group gradually became smaller as the concentration of glyphosate treatment increased, compared to the control group, while the color of the bracts in the 30 g/L treated group changed from green to light pink (Figure 1A). The anthers gradually did not dehisce, the filaments gradually shortened, the anther gradually crinkled and became smaller, and the stigma gradually elongated (Figure 1B). The treatment of 30 g/L glyphosate led to the abnormal development of ovules (Figure 1C). The peak flowering stage appeared 30 days after the glyphosate treatment, the flowers from the control group (CK) were selected to pollinate the treatment groups, and 30 flowers were hybridized in each treatment. The number of adult bolls at harvest was investigated. We found that up to 80% of the boll setting percentage was achieved in the treatment groups at concentrations below 15 g/L, while the boll setting percentage was below 6.6% at 30 g/L (Figure 1D). According to the phenotype (Figure 1E) at harvest and the above results, it recommends that when spraying the herbicide glyphosate, the maximum concentration should not exceed 15 g/L.

### 2.2. Glyphosate Attracts Male Sterility by Affecting the Shikimic Acid Pathway in Sea Island Cotton

At 15 days after glyphosate treatment, the levels of shikimic acid in the leaves gradually enhanced as the concentrations increased. Moreover, the content increased by approximately 2, 2, 3, and 5.59-fold after 15 days of treatment with 3.75, 7.5, 15, and 30 g/L of glyphosate, respectively, compared to the day of treatment (Figure 2A). In addition, the 5-enolpyruvate manganate 3-phosphate synthase (*EPSPS*) concentration in leaves also increased in parallel with the increasing glyphosate treatment concentrations. Furthermore, the 3.75 g/L treatment *EPSPS* synthase concentration, which was significantly higher than the pretreatment concentrations, and the treatments of 7.5 g/L and above, all showed significant increases compared to the pretreatments (Figure 2B).

We found that the expression of the *EPSPS* gene was significantly upregulated in the leaves after glyphosate treatment (Figure 2C), therefore, we hypothesized that glyphosate would induce the accumulation of the *EPSPS* protein by increasing the expression of the *EPSPS* gene. When the glyphosate concentration was below 7.5 g/L, the manganiferous acid pathway produced a lesser effect, yet above 7.5 g/L, the expression of the *EPSPS* gene was significantly upregulated. When the concentration was higher than 15 g/L, shikimic acid accumulated in large quantities and caused irreversible damage to the plants’ growth.

Subsequently, we analyzed the shikimic acid content, the *EPSPS* synthase concentration, and the *EPSPS* gene expression levels in different tissues after 15 days of treatment with 15 g/L glyphosate. Shikimic acid levels were the lowest in the ovaries and higher in the young buds, anthers, and leaves after glyphosate treatment. They were 2.26, 1.32, 0.82, 0.78, 0.69, 0.51, and 0.32 times higher in the leaves, anthers, young buds, stem tips, bracts, ovaries, and petals, respectively, than in the controls. The maximum difference was observed in the leaves and anthers before and after the glyphosate treatment (Figure 2D).

In addition, glyphosate treatment increased *EPSPS* synthase concentrations in the stem tips, bracts, ovaries, young buds, anthers, leaves, and petals by 1.19, 1.14, 0.83, 0.61, 0.30, 0.19, and 0.05-fold, respectively, compared to the pretreatment, with the lowest *EPSPS* concentrations observed in the anthers (Figure 2E). After glyphosate treatment, the *EPSPS* genes expression levels were 13.61, 4.10, 2.59, 2.59, 2.35, and 1.75 times higher in the ovaries, bracts, leaves, stem tips, anthers, and young buds, respectively, than in the controls (Figure 2F). The highest and the lowest were observed in the ovaries and anthers, respectively (Figure 2F). Simply, 15 g/L glyphosate causes male abortion, yet the female organs were shown to regularly grow (Figure 2G).

### 2.3. Tetrads Stage Is the Key Period during Which 15 g/L Glyphosate Affects Male Sterility in Sea Island Cotton

In the control group sprayed with water for 10 days, the anthers changed from white to yellow when the young buds were 9–10 mm in length (Figure 3A). After treatment with 15 g/L glyphosate, the anthers changed from white to yellow and the buds were 11–12 mm in length (Figure 3B), suggesting that glyphosate delays the development of the male organs in Sea Island cotton.

Then, we observed anther development and discovered that, after 5 days of treatment with 15 g/L glyphosate, the anthers dehisced normally (Figure 4A); after 10 days of treatment, about half of the anthers dehisced (Figure 4B); after 15 days of treatment, the anthers did not dehisce (Figure 4C); after 20 days of treatment, the anthers did not dehisce and no pollen grains were dispersed, whereas the pollen grains in the control group were full and dispersed normally (Figure 4D,E). In addition, we observed pollen viability and found that 3.96% of the pollen grains were abortive after 3 days of treatment (Figure 4F). The proportion of abortive pollen rose to 14.01% (Figure 4G), 55.16% (Figure 4H), 64.80% (Figure 4I), 89.36% (Figure 4J), and 97.90% (Figure 4K) after 5, 6, 7, 8 and 9 days of treatment, respectively. After spraying 15 g/L glyphosate, the flower buds previously 8–9 mm in length exhibited a completely aborted phenotype, after 20 days.

To further elucidate the critical period during which glyphosate affects male sterility, we observed microscopic sections of the anthers from 2 mm to 24 mm buds. The developmental dynamics did not differ significantly in the treatments and controls at the meiosis stage (corresponding to 7–8 mm buds) and earlier periods (<7 mm buds) (Figure 5A–F,a–f). The 8–9 mm bud corresponded to the tetrad stage when the tetrads came into being and the middle layer was squeezed; however, in the 15 g/L glyphosate treatment group, the tapetum of anther prematurely separated from the tetrads and broke up (Figure 5h). In the 9–10 mm bud, the tetrad stage anthers were further differentiated, with a tetrad shape appearing in addition to the cruciform shape, and the callus wall around the tetrad began to disintegrate, yet in the 15 g/L glyphosate-treated group, the space in the tapetal cells in the anther became larger from the tetrad stage, resulting in a block on the supply of nutrients to the microspores (Figure 5i). In the 10–11 mm bud, which corresponds to the microspore release stage, the microspores were released from the tetrads, the callus walls in the tetrad stage anther disintegrated, and the microspores gradually separated. However, the tetrads in the anthers treated with 15 g/L glyphosate were unable to dissociate into individual microspores (Figure 5j). In the 11–12 mm bud, namely the anther development stage, the anthers continued to grow and increase, the microspores formed the outer walls and vacuoles, and the tapetum began to degrade. However, the microspore morphology was abnormal in the 15 g/L glyphosate-treated group (Figure 5k). In the 12–13 mm buds, which correspond to the pollen grain development stage and its entrance into the mononuclear leaning stage, the 15 g/L glyphosate-treated microspores had an irregular morphology (Figure 5l). The 13–15 mm buds correspond to the tapetum disintegration period, for the 15 g/L glyphosate-treated anther, the pollen grains were deformed and had less content (Figure 5m). In the 15–17 mm buds, the microspore underwent mitosis for the first time, yet those treated with 15 g/L glyphosate were almost completely devoid of contents. In the 17–24 mm buds, the tapetum was thoroughly degraded, while the pollen grain tended to mature, whereas, in the 15 g/L glyphosate-treated anther, the microspores were crumpled and deformed (Figure 5n). Thus, the critical period where the anther development was affected by the 15 g/L glyphosate relates to the tetrad formation and microspore development phase, i.e., the 8–9 mm buds. Afterward, the microspore separation was abnormal and did not receive a sufficient supply of nutrients, which caused the microspore cavity to subsequently deform, which prevented the anthers from being properly decomposed.

### 2.4. Transcriptome Reveals Genes Related to Male Sterility Evoked by Glyphosate in Sea Island Cotton

To explore the mechanism through which glyphosate causes male sterility, the transcriptomes of the anthers from the 7–12 mm buds treated with water and 15 g/L glyphosate were sequenced. From these, we obtained a total of 79,184 transcripts including 19,683 known transcripts and 59,501 unknown transcripts (Appendix A). The majority of the transcripts were specific to each developmental period of the anther, with the highest number of upregulated (479) and downregulated (494) genes in the 8–9 and 11–12 mm buds, respectively (Figure 6A,B and Appendix A). We used qRT-PCR to validate the expressions of 15 genes associated with fertility (Appendix A) and found that the expression trends of the genes in the transcriptome and qRT-PCR data were consistent, with an overall correlation of 0.76 (Appendix A), thereby demonstrating the reliability of the transcriptome data. In the critical period of glyphosate-induced male sterility, namely the tetrad formation phase (8–9 mm buds), GO (Gene Ontology) enrichment analysis revealed that the biological process annotated to the largest number of genes was the abscisic acid response; the cellular components were nuclear and Golgi, and the molecular functions were protein binding and protein dimerization activity (Figure 6C,D). In addition, KEGG (Kyoto Encyclopedia of Genes and Genomes) enrichment analysis of the differential genes during the tetrad formation (8–9 mm) identified the top three metabolic pathways: biosynthesis of secondary metabolites, and phytohormone signaling (Figure 6E,F). Thus, we can conclude that phytohormone signaling, especially the abscisic acid response, is one of the pathways responsible for glyphosate-induced male sterility.

Subsequently, we found that after 15 g/L of glyphosate treatment, abscisic acid content significantly increased in the anthers of the 7–12 mm buds. The treated group had a 37.83%, 25.70%, 25.70%, 25.70%, 25.82%, and 13.71% higher ABA (Abscisic acid) content than the control group in the anthers of the 7–8 mm, 8–9 mm, 9–10 mm, 10–11 mm, and 11–12 mm buds, respectively (Figure 7A). Next, we analyzed the expression differences in the abscisic acid-responsive genes in the anthers of the 8–9 mm buds (abortive critical period) after 15 g/L glyphosate treatment (Figure 7B). We found a significant increase in the expression of a key gene, *Gbar_A11G003090*, after glyphosate treatment. The gene encodes the *TCP14* transcription factor, a member of the TCP gene family that is reportedly linked to fertility. We performed qRT-PCR on this gene to verify its expression and the results showed that the gene expression was significantly upregulated after glyphosate treatment (Figure 7C). Promoter analysis of the gene revealed the presence of multiple species of cis-acting elements in the gene, including two responsive elements with an abscisic acid response (Figure 7D). As such, it could be a key candidate gene for future follow-up studies on glyphosate-induced male sterility in Sea Island cotton.

## 3. Discussion

### 3.1. Glyphosate Triggered Male Sterility in Cotton

Anthers are the male organs in plants that dehisce at maturity to disperse pollen to the stigma, a key process in plant fertilization [24]. Yasuor et al. suggested that glyphosate hindered anther dehiscence in cotton, which correlates with temperature and auxin accumulation [25]. Pline et al. found that the stigma of glyphosate-treated plants was 1.2–1.4 mm longer and the anther filaments were 0.8–0.9 mm shorter than the controls [26]. The filaments play a role in transporting nutrients to the anthers. The filament length determines the distance between the anthers and the stigma and plays a key role in the pollination process. Filament fusion facilitates the fixation of the stamens around the stigma, which in turn supports pollination and protects the ovaries [27]. In the present study, glyphosate inhibited anther dehiscence and broadened the distance between the stamens and stigma, and shortened the filaments, which hints at a mechanism for the glyphosate-induced male sterility in Sea Island cotton.

### 3.2. The Critical Periods for the Emergence of Male Sterility

Male sterility occurs at various periods in plants. The problematic tapetum was the main cause of abnormal microspore development in wheat [28]. In rice, male sterility was caused by the failure of tapetum degeneration, resulting in a nutrient deficiency of the microspores, and ultimately, pollen abortion [29]. In lilies, abortive microspores formed tetrads surrounded by undegraded calluses in late meiosis, impeding the exchange of energy and nutrients from the surroundings to the pollen grains, which may eventually lead to pollen abortion [30]. In the present study, we also discovered incomplete dissociation of the microspores. Pollen abortion occurred between spore formation and the tetrad stage in most dicotyledons [31]. Here, the critical period for glyphosate-induced male abortion was the tetrad stage in the dicotyledonous Sea Island cotton. The incomplete degradation of the calluses and the premature degradation of the tapetum cells resulted in insufficient nutrient sources for microspore development and prevented the microspore from developing into an active pollen grain.

### 3.3. Abnormal Abscisic Acid Pathway Is Responsible for Male Sterility

A large number of genes have been identified in plants, e.g., 3500 genes are expressed in Arabidopsis anthers [32]; 1742 differential genes have been identified in cotton [33]; there are 2446 differential genes in 1355A sterile lines, compared to fertile lines [34]. In total, 2795 differential genes were identified in this study; GO and KEGG enrichment analyses revealed that phytohormone-related pathways were heavily enriched, particularly the abscisic acid response pathway, while abscisic acid levels increased significantly after glyphosate treatment, suggesting that glyphosate affects male fertility by modulating the abscisic acid pathway in Sea Island cotton. In barley, abscisic acid guarantees programmed cell death, which is a prerequisite for male gametophyte development [35]. In Arabidopsis, insensitivity to abscisic acid could lead to male abortion [36]. In rice, abscisic acid-induced male sterility by regulating sugar transport in the apoplast [37]. In oilseed rape, the abscisic acid was involved in tapetum dissociation in the early anther development stages, which is strongly associated with male sterility [38].

Some genes that indirectly affect abscisic acid might also contribute to male sterility. In citrus, the transcription factor, *CsHB5*, promoted apoptosis by regulating abscisic acid biosynthesis genes [39]. In Arabidopsis, the transcription factor *CDF4* promotes floral organ abscission by regulating abscisic acid pathways [40]. In the present study, some abscisic acid response genes were identified, such as *Gbar_D07G022720*, which is homologous to a zinc finger protein transcription factor that regulates the abscisic acid response in Arabidopsis [41]. Another candidate gene, *Gbar_A05G033650*, is the homolog of *AT3G08550*, which is involved in the abscisic acid signaling pathway, cell wall biosynthesis, and degradation [42]; its function loss in Arabidopsis exhibits cell wall defects, impaired production of lignin and cellulose, and callose accumulation [43]. In the present study, similar phenomena were found, such as the incomplete degradation of the callus during tetrad separation.

## 4. Materials and Methods

### 4.1. Plant Material

The plant material used in this experiment was Sea Island cotton (*Gossypium barbadense*) Xinchang 5, a genetic-stable *CP4-EPSPS* (encoding 5-enolpyruvic acid manganese-3-phosphate synthase *EPSPS* from *Agrobacterium rhizogenes* strain *CP4*) transgenic long-staple cotton line. This gene was successfully cloned by PCR using primers designed from the *CP4-EPSPS* gene sequence released by the Chinese Ministry of Agriculture (Figure 1). Xinchang 5 is intended for scientific testing only and has not been approved for variety or event use. The material is tolerant to 3 g/L glyphosate at the seedling stage while spraying it at the bud stage can cause sterility.

### 4.2. Experimental Treatment

The trials were conducted in 2021 and 2022 at the Xinjiang Academy of Agricultural Sciences experimental base in Korla (85°85′53″ E, 41°68′84″ N). The soil is loam in texture. Five treatments with three replications were established in a 1-membrane/4-row (10 + 66 + 10 cm) machine-harvesting pattern. Glyphosate was sprayed at the mid-bud stage (10-leaf stage). The concentrations applied were 3.75 g/L, 7.5 g/L, 15 g/L, and 30 g/L. A water control (CK) was also used. In the cotton fields, water and manure were managed in a regular manner.

### 4.3. Sampling Methods

Pollens sprayed with 15 g/L glyphosate and water were taken daily for 4% TTC staining [44] and observed under a stereomicroscope. Fresh young leaves from the different treatments and controls were frozen in liquid nitrogen 15 days before and after treatments and stored in a low-temperature refrigerator at −80 °C. In 2022, 2–3, 3–4, 4–5, 5–6, 6–7, 7–8, 8–9, 9–10, 10–11, 11–12, 12–13, 13–14, 14–15, 15–16, 16–17, 17–18, 18–19, 19–20, 20–21, 21–22, 22–23, and 23–24 mm buds were taken from the treated and control groups. The anthers were used for fixation and later for paraffin sections. The bracts, sepals, and petals were stripped, the pure anthers were collected, and the stigma, ovaries, and filaments were discarded. The anthers were quickly separated at low temperatures and frozen in liquid nitrogen, after which they were stored in a low-temperature refrigerator at −80 °C until use. Furthermore, fresh young buds, ovaries, leaves, anthers, petals, bracts, and stem tips were taken 15 days after treatment and stored as above.

### 4.4. Detection of EPSPS Gene Expression Levels

Fresh young buds, ovaries, leaves, anthers, petals, bracts, and stem tips were taken from the glyphosate-treated groups for 15 days, and three replicates were taken from each sample. All samples were ground in liquid nitrogen. Total RNA was extracted using an RNAprep pure Micro Kit (TIANGEN, Beijing, China). RNA was reverse-transcribed using the HyperScriptTM Ⅲ RT SuperMix for qPCR with gDNA Remover (NovaBio, Shanghai, China). Moreover, qRT-PCR was performed using a 2X S6 Universal SYBR qPCR Mix (NovaBio, Shanghai, China). The PCR steps included a 95 °C pre-denaturing step for 30 s, 95 °C denaturation for 10 s, 60 °C annealing and extension for 30 s, a total of 45 cycles were included. The primer sequences are shown in Appendix A.

### 4.5. Examination of Shikimic Acid and EPSPS Synthase Contents

After glyphosate treatment for 15 days, fresh buds, ovaries, leaves, anthers, petals, bracts, and stem tips were taken and stored in a refrigerator at −80 °C. Three replicates of 1 g each were placed in a mortar and 9 g of pH 7.2–7.4 PBS was added, and the samples were well-ground. The samples were centrifuged at room temperature for 20 min (3000 rpm/min). The supernatant was collected and assayed, according to the product manual of the ELISA Kit HCT (KANGLANG, Shanghai, China), and detected at an OD value of 450 nm. The results were calculated from the standard curve (Appendix A).

### 4.6. Histological Section Observation

To determine the critical period of pollen abortion caused by glyphosate, 2–24 mm buds were fixed in FAA (formaldehyde, acetic acid, ethanol, and water, 10:5:50:35, *v*/*v*), then, dehydrated, transparent, wax-dipped, embedded, sectioned, spread, dried, transparent, rehydrated, stained (5% toluidine blue, *v*/*v*), dehydrated, transparent, and sealed [45]. The permanently stained sections were observed and photographed using a DM2500 (Leica) microscope (Wetzlar, Germany).

### 4.7. Transcriptome Data Analysis

The RNA from the 7–8, 8–9, 9–10, 10–11, and 11–12 mm anthers treated with 15 g/L glyphosate and water was extracted using the CTAB method [46]. The 150 bp sequenced reads were obtained by library sequencing (Big, Wuhan, China). Sequencing data were analyzed using fastq [47] for quality control. The clean data were compared with the reference genome [*Gossypium barbadense* (AD_2_) ‘3–79’ genome HAU_v2_a1, https://www.cottongen.org/ (accessed on 25 April 2022)] [48], using STAR [49]. Gene quantification was performed using stringtie [50]. DESEQ2 was used to screen DEGs (differentially expressed genes) [51] with the following standards: | log2 (FoldChange) | > 2 and *p*-value < 0.05. The DEGs were used for GO (Gene Ontology) [52] and KEGG (Kyoto Encyclopedia of Genes and Genomes) [53] analyses.

### 4.8. qRT-PCR Validation

The RNA used for the transcriptome sequencing was also reverse-transcribed for real-time fluorescence quantification, according to method 4.4. There were 45 cycles of pre-denaturation at 95 °C for 30 s; denaturation at 95 °C for 10 s and annealing and extension at 60 °C for 30 s. Primers were designed using Oligo7 software and synthesized by Sangon Biotech (Shanghai, China), the primer sequences are shown in Appendix A, and the 2^−ΔΔCt^ method was used to calculate the relative gene expressions [54].

### 4.9. Phytohormone Measurement

The anthers from the same period as the transcriptome were used to measure the amount of ABA by HPLC (high-performance liquid chromatography) [55]. HPLC analysis was performed on a Shimadzu LC-20A, Shim-packGISTC18 column (250 × 4.6 mm, 5 μm; Shimadzu, Japan), eluted with HCOOH-H_2_O (1:999 (*v*/*v*), solvent A) and CH_3_OH (solvent B), respectively. The elution conditions were as follows: 0–11 min, 5–95% B; the column was drenched with 95% B and restored to the starting fraction of 5% B. The injection volume was 10 µL; the column temperature was 35 ℃. The flow rate was 1.0 mL/min, and the wavelength was set at 254 nm. These results were calculated according to the standard curve (Appendix A).

## 5. Conclusions

In this study, we analyzed the cytological and physiological properties and transcriptomic data of Sea Island cotton, before and after glyphosate treatment. The maximum concentration of the chemical herbicide was determined as 15 g/L. The critical transition period for pollen abortion in Sea Island cotton is the tetrad phase, corresponding to buds of 8–9 mm. The abscisic acid levels increased significantly after glyphosate treatment. Transcriptome analysis revealed 2795 differentially-expressed genes between the glyphosate-treated and control groups, with the highest number of genes enriched in the abscisic acid response pathway. We retrieved a significantly upregulated abscisic acid response gene, *Gbar_A11G003090*, a transcription factor, *TCP14*, whose follow-up study will be crucial in elucidating the mechanism of male sterility in Sea Island cotton caused by glyphosate. This study offers some guidance on the use of herbicides in cotton production and provides some insights into the mechanism of yield reduction due to the inappropriate use of chemical herbicides, which provides great practical reference value.

## Figures and Tables

**Figure 1 plants-12-01058-f001:**
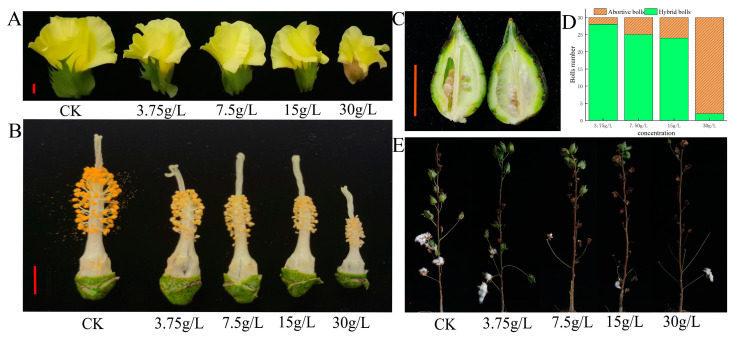
Fertility and yield-related phenotypes of Sea Island cotton after treatment with different concentrations of glyphosate. (**A**) Flower phenotypes of Sea Island cotton treated with different concentrations of glyphosate; (**B**) stigma and anther phenotypes of Sea Island cotton treated with different concentrations of glyphosate; (**C**) boll longitudinal and seed phenotypes of Sea Island cotton treated with different concentrations of glyphosate; (**D**) boll number of Sea Island cotton treated with different concentrations of glyphosate; (**E**) plant performance of Sea Island cotton treated with different concentrations of glyphosate. The scale bar is 1 cm.

**Figure 2 plants-12-01058-f002:**
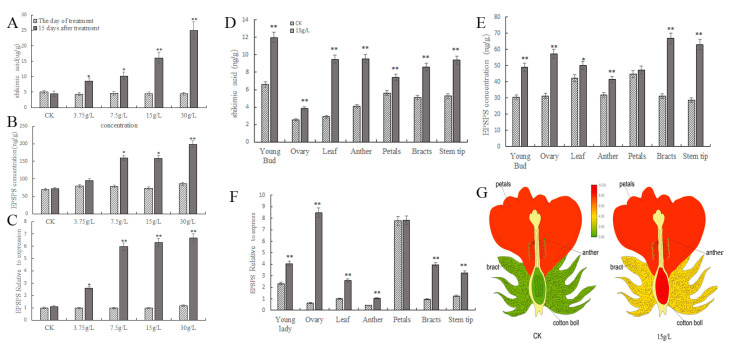
The effect of glyphosate on the shikimic acid content, *EPSPS* concentration, and *EPSPS* gene expression in different tissues. (**A**) Shikimic acid content in the leaves after treatment with different concentrations of glyphosate. * indicates a significance level of *p* < 0.05, ** indicates an extremely significant level of *p* < 0.01; (**B**) *EPSPS* synthase concentration in the leaves after treatment with different concentrations of glyphosate; (**C**) the expression levels of the *EPSPS* gene in the leaves after treatment with different concentrations of glyphosate; (**D**) shikimic acid concentration in different tissues after treatment with 15 g/L glyphosate; (**E**) *EPSPS* synthase concentration in different tissues after treatment with 15 g/L glyphosate; (**F**) differential expression of the *EPSPS* genes in different tissues after treatment with 15 g/L; (**G**) expression heatmap of the *EPSPS* gene in different tissues after treatment with 15 g/L glyphosate.

**Figure 3 plants-12-01058-f003:**
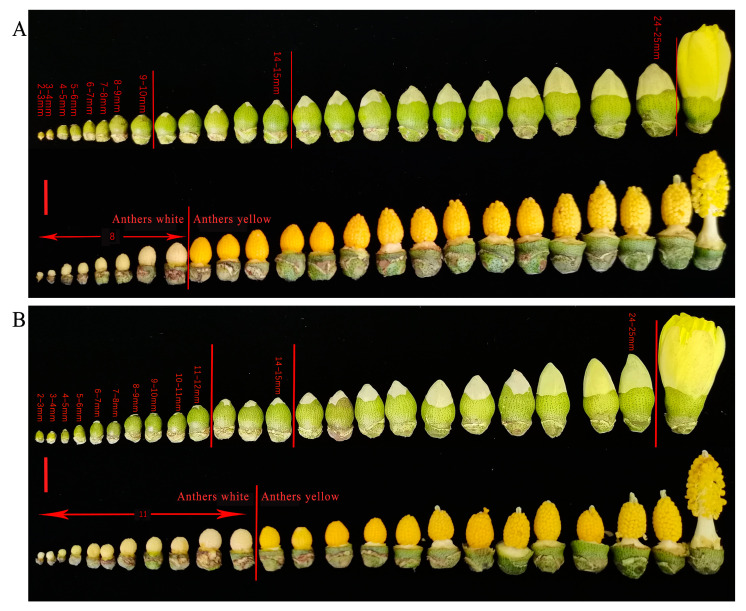
The effect of glyphosate on buds and anthers. (**A**) Developmental dynamics of buds and anthers treated with water; (**B**) developmental dynamics of buds and anthers treated with 15 g/L glyphosate. The scale bar is 1 cm.

**Figure 4 plants-12-01058-f004:**
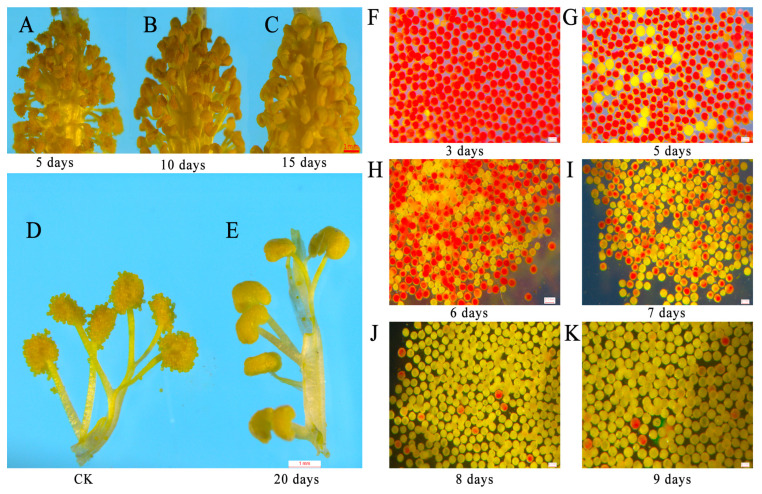
The effect of glyphosate on the anther dehiscence and pollen grain viability. (**A**–**C**) The anthers were treated with 15 g/L glyphosate for 5 days (**A**), 10 days (**B**), and 15 days (**C**); (**D**) the anthers were treated with water for 20 days; (**E**) the anthers were treated with 15 g/L glyphosate for 20 days; (**F**–**K**) the pollen grains treated with 15 g/L glyphosate for 3 days (**F**), 5 days (**G**), 6 days (**H**), 7 days (**I**), 8 days (**J**), 9 days (**K**). (**A**–**E**) The scale bars are 1 mm. (**F**–**K**) The scale bars are 0.1 mm.

**Figure 5 plants-12-01058-f005:**
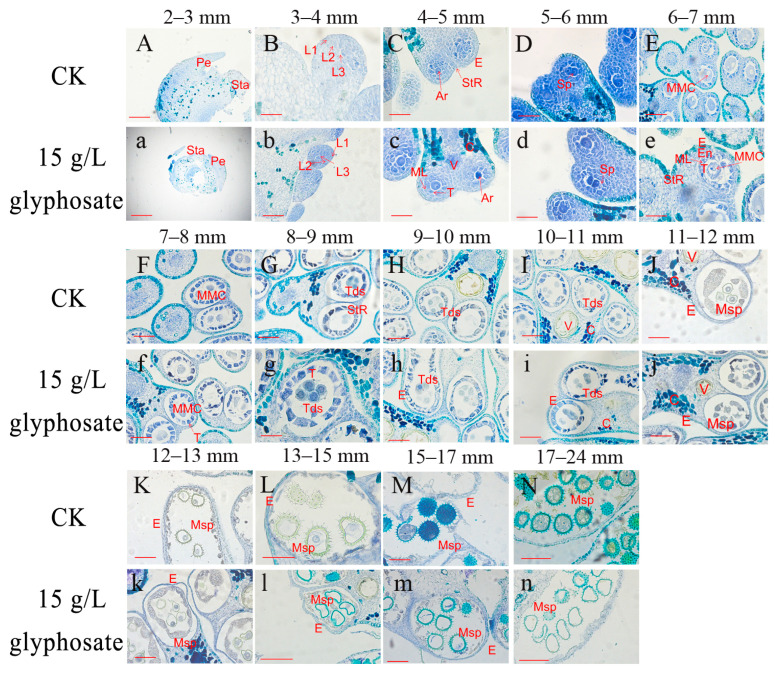
Cytological observations from glyphosate treatment on male gametophyte development. (**A**–**N** and **a**–**n**) Histological sections of anthers under CK (**A**–**N**) and 15 g/L glyphosate treatment (**a**–**n**) in 2–6 mm buds (**A**–**D** and **a**–**d**); pre-pollen mother cell development in 6–8 mm buds (**E**–**F** and **e**–**f**); meiotic separation period in 8–11 mm buds (**G**–**I** and **g**–**i**; tetrad stage), 11–17 mm buds (**J**–**M** and **j**–**m**; tapetal degradation stage), 17–24 mm buds (**N** and **n**; anther dehiscence stage). Pe: petal primordium; Sta: stamen primordium; L1, L2, and L3 are triple stamen primordia; P: periderm; Ar: sporogenic cell; Sp: sporogenic cell; E: epidermis; En: endodermis; ML: mesoderm; T: felted layer; StR: cleavage region; C: connecting zone; V: vascular zone; St: cleft; MMC: pollen mother cell; Tds: tetrad; Msp: microspore. Msp: microspore. a: the scale is 200 µm, and in the other figures, it is 100 µm.

**Figure 6 plants-12-01058-f006:**
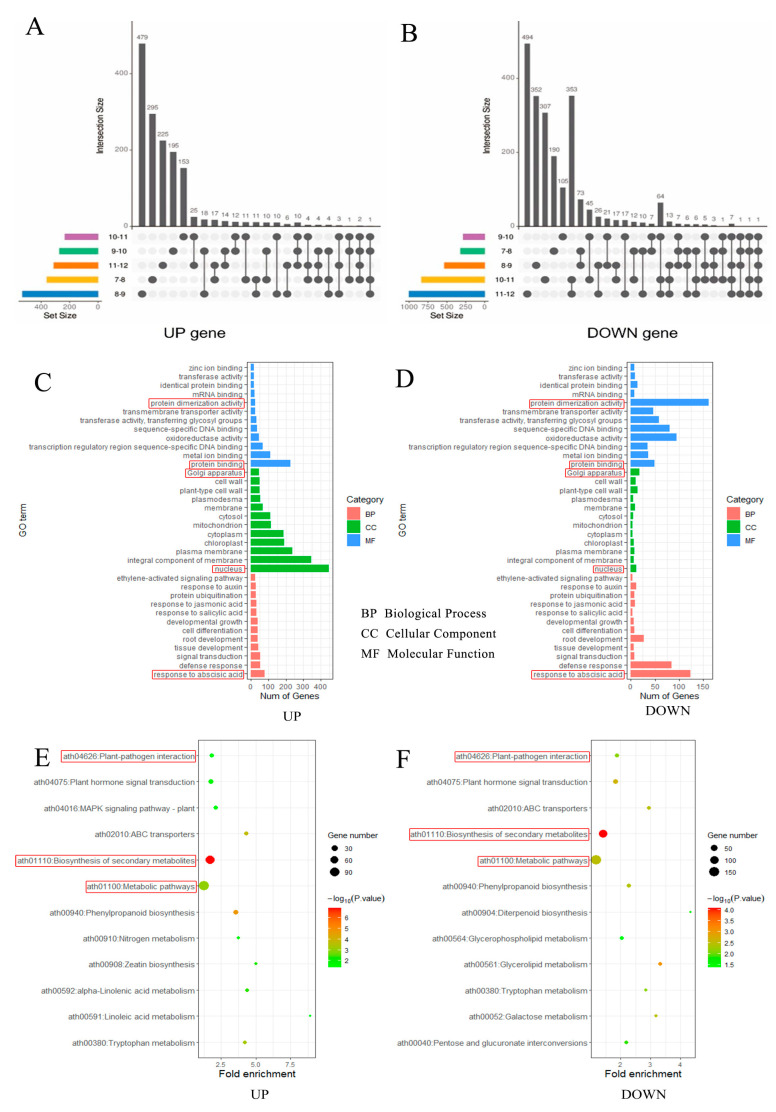
The number, GO annotation, and KEGG enrichment of differentially expressed genes were identified in glyphosate-treated and control groups. (**A**,**B**) The number of differentially expressed genes in the anthers of different lengths of buds; (**C**,**D**) GO annotation of the differentially expressed genes; (**E**,**F**) KEGG enrichment of the differentially expressed genes.

**Figure 7 plants-12-01058-f007:**
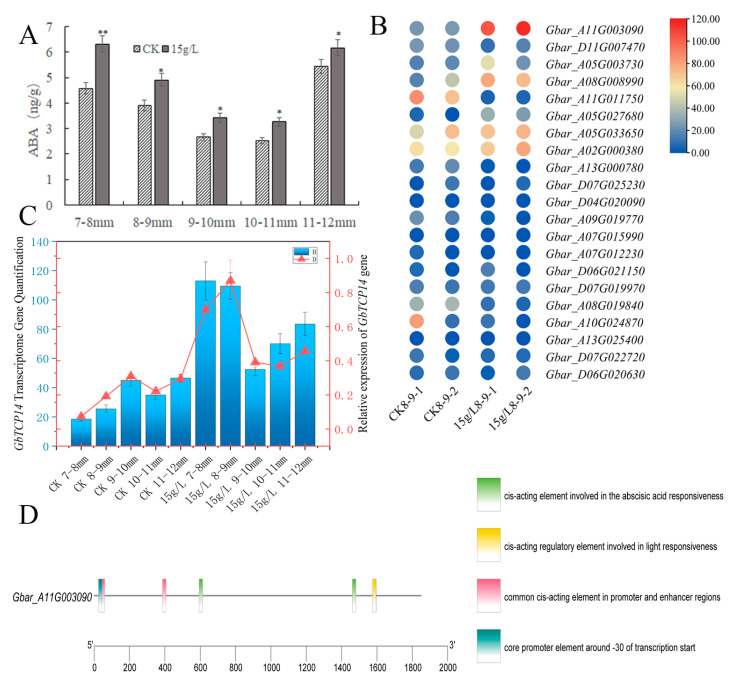
Abscisic acid content in anthers and the expression of abscisic acid response-related genes in glyphosate-treated and control groups. (**A**) Abscisic acid content in anthers of different-lengthed buds. * indicates a significance level of *p* < 0.05 and ** indicates an extremely significant level of *p* < 0.01. (**B**) Expression heatmap of the abscisic acid response-related genes in anthers of 8–9 mm buds. (**C**) Expression of *GbTCP14* gene in transcriptome with qRT–PCR. (**D**) Promoter analysis of the *Gbar_A11G003090* gene.

## Data Availability

All data generated or analyzed during this study are included in this published article and information files. The raw sequencing data used during this study have been deposited in the NCBI’s SRA with the accession number PRJNA929336 (https://www.ncbi.nlm.nih.gov/sra/PRJNA929336 (accessed on 23 February 2023).

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
