# Peer review of "Glyphosate-Induced Abscisic Acid Accumulation Causes Male Sterility in Sea Island Cotton"

_plants, 2023, doi:10.3390/plants12051058_

Round 1
Reviewer 1 Report
I have received the Manuscript (ID: plants-2247936) entitled: ‘Glyphosate-induced abscisic acid accumulation causes male sterility in Sea Island cotton’ submitted to the Plants for a review. The topic of the submitted Manuscript is highly interesting to the academic community because it describes in a clear and coherent way the research involving male sterility induced by glyphosate in sea island cotton. The quality of the descriptions and utilized methodologies is excellent. I also paid attention to the exceptionally good quality and readability of the graphics in the document, which proves the high scientific professionalism of authors. All sections, including the data acquisition and analysis as well as writing the discussion, are clear and supported with appropriate literature reports. Therefore, I strongly recommend publication of the submitted Manuscript in Plants after correction of minor errors/ambiguities:
· Line 15: the term “promotion of pollen defeat” is a bit unclear and I recommend to transform it into more international phrase, e.g. "pollen abortion" is much more recognizable
· Line 49: Here “[9].” instead of dot there should be coma
· Line 121-124: the sencence: “The shikimic acid pathway is normal at concentrations below 7.5 g/L, but the concentration of glyphosate at and above was too high, while over 15 g/L, the precursor of shikimic acid precursor will increase substantially, causing irreversible damage to plant growth.” is confusing. I propose to divide it into 2 separate sentences
· Line 180: “observed microscopic sections of the anthers from 2 mm to 24 mm buds” – can you provide more details regarding microscope used and the observation in the methodology?
Reviewer 2 Report
This article shows careful and rigorous study and interesting scientific phenomena regarding the topic.
General concern: The variety of Sea Island cotton used in the study, cv. Xinchang 5, is transgenic for expression of CP4-EPSPS to induce tolerance to glyphosate. The study focuses on effects of glyphosate on aba accumulation leading to male sterility. In 4.1 'Plant material' section, is it possible to comment on the specific insertion event of this variety and its efficacy and tolerance compared to known commercial events. Different transgenic events will show different levels of tolerance to herbicide, and it would be helpful to understand the level of tolerance to glyphosate in this particular event. For example, in upland cotton, commercial event MON-1445 (Roundup Ready) could not be sprayed over the top after 4th true leaf or these sterility effects would occur, whereas event MON88913 (Roundup Ready Flex) could tolerate glyphosate treatment later in development without those effects. Is there a reference that can be used for either the variety registration or the event approval?
Specific suggestions: Reference #3 is cited following the statement "Chemical weed control is currently one of the most common methods in the field." While this introductory statement is true, the citation is from 1968 and likely does not include more current use of herbicides, especially since late 1990 introduction of transgenic glyphosate-tolerant row crops. Recommend citing a more recent article for the statement.
Line 44 - change Ryan et al. to Viator et al. (use family name instead of given name.
Line 47 - change Hagai et al. to Yasuor et al.
Line 49 - add 'in yellow nutsedge' after 'content of carotenoid.'
Line 66 - add 'in Japanese apricot ' after 'causes pistil abortion.'
Line 92 - change 'full-bloom stage' to 'peak flowering stage.'
Line 238 - remove duplicate 'metabolic pathways.'
Line 274 - change 'Wendy et al.' to 'Pline et al.'
Line 285 - Reference 28 refers to wheat, not sunflower.
Section 4.1 - would be helpful to repeat species (Gossypium barbadense) in the plant material description.
Nice work, though the subject matter does not appear well aligned with the special issue to which it is submitted.
Author Response
Please see the attchment
